# Comparison of geological clusters between influenza and COVID-19 in Thailand with unsupervised clustering analysis

**Thanin Methiyothin[1,2], Insung Ahn[1,2]***

1 Department of Data-Centric Problem Solving Research, Infectious Disease AI Team, Korea Institute of Science and Technology Information, Yuseong-gu, Daejeon, Republic of Korea, 2 Department of Applied AI, University of Science & Technology, Yuseong-gu, Daejeon, Republic of Korea

* isahn@kisti.re.kr

**Data Availability Statement:** All FluNet's influenza data are available from the URL (https://www.who.int/tools/flunet/). All influenza surveillance data file from the Department of Disease Control (DDC), under the Thai Ministry of Public Health are

## Abstract

The coronavirus disease (COVID-19) pandemic has considerably impacted public health, including the transmission patterns of other respiratory pathogens, such as the 2009 pandemic influenza (H1N1). COVID-19 and influenza are both respiratory infections that started with a lack of vaccination-based immunity in the population. However, vaccinations have been administered over time, resulting in a transition of the status of both diseases from a pandemic to an endemic. In this study, unsupervised clustering techniques were used to identify clusters of disease trends in Thailand. The analysis incorporated three distinct surveillance datasets: the pandemic influenza outbreak, influenza in the endemic stage, and the early stages of COVID-19. The analysis demonstrated a significant difference in the distribution of provinces between Cluster -1, representing those with unique transmission patterns, and the other clusters, indicating provinces with similar transmission patterns among their members. Specifically, for Pandemic Influenza, the ratio was 61:16, while for Pandemic COVID-19, it was 65:12. In contrast, Endemic Influenza exhibited a ratio of 46:31, with a notable emergence of more clustered provinces in the southern, western, and central regions. Furthermore, a pair of provinces with highly similar spreading patterns were identified during the pandemic stages of both influenza and COVID-19. Although the similarity decreased slightly for endemic influenza, they still belonged to the same cluster. Our objective was to identify the transmission patterns of influenza and COVID-19, with the aim of providing quantitative and spatial information to aid public health management in preparing for future pandemics or transitioning into an endemic phase.

## Introduction

Since its worldwide outbreak in early 2020, the coronavirus disease (COVID-19) pandemic has brought about significant changes in nearly every aspect of public health. Many countries have adopted non-pharmacological interventions (NPIs), including the utilization of masks, self-isolation, quarantine measures, and territorial or community-wide lockdowns, as

available from the URL (http://doe.moph.go.th/surdata/disease.php?dcontent=old&ds=15/). All COVID-19 surveillance data file from the Department of Disease Control (DDC), under the Thai Ministry of Public Health are available from the URL (https://covid19.ddc.moph.go.th/). All population statistics data file from the Department of Provincial Administration, under the Thai Ministry of Interior are available from the URL (https://stat.bora.dopa.go.th/new_stat/webPage/statByYear.php). All passenger traffic data from the International Air Transport Association (IATA) cannot be shared publicly because of charged service. Data are available from conclusion of agreement for data usage with the International Air Transport Association (IATA). The data underlying the results presented in the study are available by login from the url (https://www.iata.org/).

**Funding:** This research was financially supported by the Ministry of Science and ICT, Republic of Korea in the form of a grant (K-23-L04-C06-S01) received by IA and TM. This research was also financially supported by the Government-wide R&D Fund project for infectious disease research (GFID), Republic of Korea in the form of a grant (HG23C1624) received by IA and TM. The funders had no role in study design, data collection and analysis, decision to publish, or preparation of the manuscript.

**Competing interests:** The authors have declared that no competing interests exist.

strategies to mitigate human-to-human transmission. As of August 21, 2023, there have been over 770 million confirmed cases and more than 6.9 million reported fatalities [1–3]. A similar scenario can be observed in the 2009 influenza pandemic (H1N1), the World Health Organization (WHO) announced the first human influenza pandemic in the 21st century in June 2009. This novel virus emerged from a triple-reassortant North American swine influenza A virus that acquired two viral genes from the Eurasian swine influenza A virus. The outbreak originated in Mexico in March 2009. By the end of July 2009, confirmed cases of the H1N1 2009 pandemic influenza were reported in over 168 countries, with the number of laboratory-confirmed cases exceeding 162,380, resulting in 1,154 fatalities. Notably, pregnant women, individuals with underlying medical conditions, those who were morbidly obese, and children under 2 years old have been the highest risk of severe illness during this pandemic influenza. Moreover, data from various regions worldwide concerning hospital and intensive care unit (ICU) admissions revealed that one-third of patients experiencing severe illness were healthy adults [4].

Both COVID-19 and influenza are respiratory infections that share a similarity in that when their outbreaks initially started, the population did not have immunity through vaccination. However, as time progressed, the population began to receive vaccinations, leading to a transition of the status of both diseases from pandemic to endemic [2,4,5]. Therefore, this study aims to analyze surveillance data for COVID-19 alongside influenza, providing insights into the progression from a pandemic to an endemic phase in the realm of respiratory infectious diseases. Furthermore, understanding the transmission patterns of respiratory pathogens like influenza and COVID-19 pandemic is a fundamental aspect of effective public health management and preparedness. This knowledge enables authorities to develop and implement evidence-based strategies to control the spread of diseases, reduce their impact, and ultimately protect public health. It also lays the groundwork for a more proactive and resilient response to future pandemics.

Owing to the emergence of the COVID-19 pandemic, the influenza season in 2020–2022 was unusual. The implementation of public health measures, such as social distancing, mask-wearing, and increased hand hygiene, helped mitigate the spread of both influenza and COVID-19. The number of influenza cases and associated hospitalizations were lower in the United States and globally [1,6]. Furthermore, Mexico observed a significant decrease in influenza cases following the implementation of public health measures aimed at controlling COVID-19. The containment strategies designed to minimize the spread of the new respiratory disease also proved to be highly effective in reducing influenza cases [7]. Much like the experience in Singapore during 2020, the adoption of health measures to limit the transmission of viral respiratory diseases had the potential to mitigate the impact of the COVID-19 pandemic and lead to a significant reduction in cases of influenza-like illness [8]. According to influenza data reported to the World Health Organization's (WHO) FluNet, the number of infected cases in the United States decreased from 164,538 in 2019 to 105,921 and 19,735 in 2020 and 2021, respectively. In South Korea, the number of infections decreased from 2,712 in 2019 to 992 and zero in 2020 and 2021, respectively. In Thailand, influenza cases decreased from 3,078 in 2019 to 590 and 46 in 2020 and 2021, respectively [9]. Table 1 shows the

**Table 1. Incidence of influenza infections in the United States, South Korea, and Thailand.**

| Country/Year | 2019 | 2020 | 2021 |
|---|---|---|---|
| United States | 164,538 | 105,921 | 19,735 |
| South Korea | 2,712 | 992 | 0 |
| Thailand | 3,078 | 590 | 46 |

incidence of influenza infections in the United States, South Korea, and Thailand between 2019 and 2021.

Influenza is transmitted through person-to-person contact by means of aerosols that are released during coughing, sneezing, and breathing. Seasonal epidemic patterns of influenza are complicated and include population levels, environmental factors, human crowding, rapid viral evolution, and international travel patterns [10]. In temperate regions, influenza outbreaks are strongly influenced by seasonal fluctuations in temperature and absolute humidity. However, tropical countries experience less pronounced annual climate cycles, resulting in influenza outbreaks that exhibit reduced seasonality and are more challenging to explain through environmental correlations [11].

In the case of COVID-19, transmission can occur through direct means, such as droplet transmission and human-to-human contact, as well as indirect contact with contaminated objects and airborne contagion [12]. The first recorded case of COVID-19 in Thailand was a resident of Wuhan who arrived in Bangkok on January 8, 2020. This case marked the first instance of COVID-19 transmission outside China [13]. The first COVID-19 case in Thailand highlighted the impact of international travel patterns on viral transmission. This is evident from the analysis of air travel passenger traffic data provided by the International Air Transport Association (IATA) [14]. The data revealed that, in 2019, the highest number of incoming passengers to Thailand was from China, reflecting the air transport situation prior to the global COVID-19 pandemic. This finding is graphically represented in Fig 1.

Owing to the complex nature of the spread patterns of COVID-19 and influenza, this study employed clustering techniques to identify clusters of disease trends in Thailand. The analysis incorporated three distinct surveillance datasets: influenza surveillance data from 2009–2011 (referred to as Pandemic Influenza), representing the initial stage of the pandemic influenza outbreak; influenza surveillance data from 2017–2019 (referred to as Endemic Influenza), reflecting the endemic status and population immunity resulting from vaccination; and COVID-19 surveillance data from 2020–2022, representing the early stages of the COVID-19 situation (referred to as Pandemic COVID-19). The datasets were obtained from the Department of Disease Control, Ministry of Public Health, Thailand [15,16].

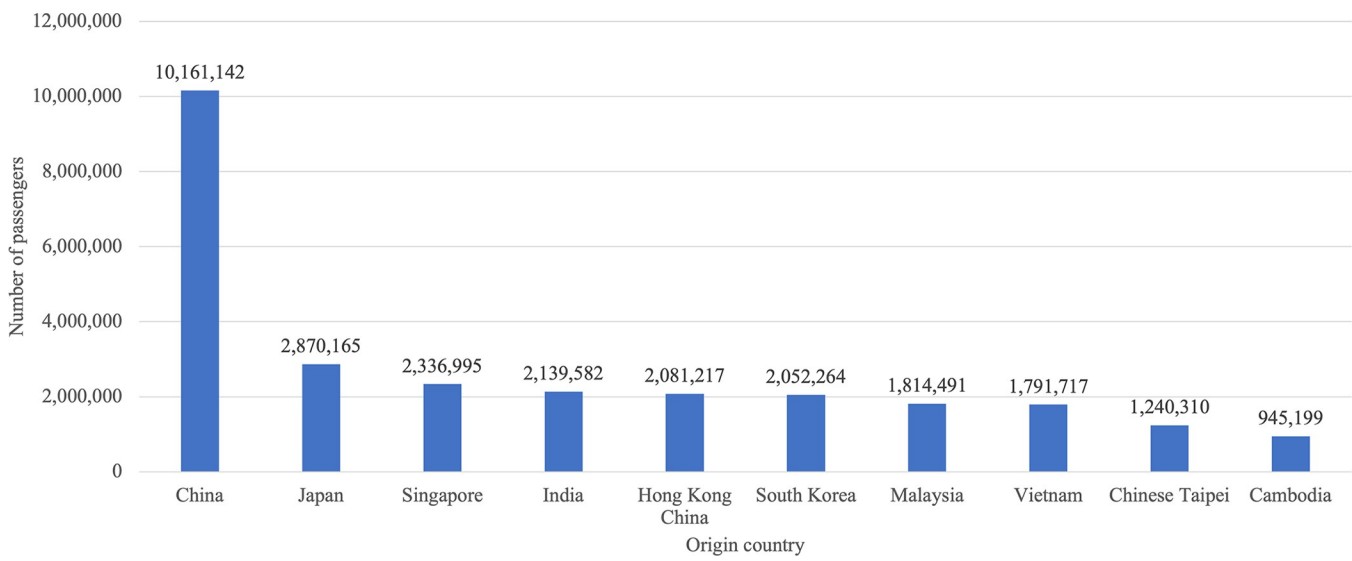

**Fig 1. The total number of passengers traveling to Thailand by airplane in 2019, categorized by country.**

This study aimed to analyze the trends in Pandemic Influenza, Endemic Influenza, and Pandemic COVID-19 in Thailand. The goal was to identify the spread patterns in regional and provincial levels, both during the early stages of the pandemic and after transitioning to an endemic phase. Our findings revealed a greater similarity in the number of provinces exhibiting unique transmission patterns between Pandemic Influenza and Pandemic COVID-19 when compared to Endemic Influenza. Additionally, clusters of provinces sharing similar transmission patterns were generally located in the northeastern region. However, following the transition of influenza to an endemic state, more clustered provinces emerged in the southern, western, and central region. This study aims to offer valuable insights for public health management, enabling more effective responses to newly emerging pandemics and endemic situations at both the national and regional levels. By prioritizing appropriate measures and resources, this analysis can contribute to the improvement of public health strategies.

## Method

### Data collection

The Department of Disease Control (DDC), under the Thai Ministry of Public Health (MoPH), is responsible for monitoring, controlling, and preventing the spread of infectious diseases in Thailand. The DDC collects, analyzes, and disseminates data on infectious diseases in Thailand. The DDC maintains a comprehensive information database on infectious diseases in Thailand, including data on disease incidence, prevalence, and mortality. As an infectious-disease dataset provider, the DDC provides a range of data to the public. This includes data on the incidence and prevalence of infectious diseases in Thailand, as well as information on disease outbreaks, disease surveillance, and other related topics [15,16]. We obtained comprehensive surveillance data for COVID-19 and influenza from the Department of Disease Control (DDC), specifically from each province, in RTF format for each year. Using Python, we extracted input features such as disease name, year, month, city, and the number of cases. These data were then organized into a MySQL database for subsequent analysis. The dataset consists of 17,460 records of monthly influenza surveillance data spanning from January 2003 to December 2021. Additionally, there are 11,858 records of weekly COVID-19 surveillance data covering the period from week 3 of 2020 to week 52 of 2022, in Excel format. Using Python, we extracted input features including year, week, city, and the number of cases, and similarly stored this data in a MySQL database for further analysis.

We acquired national population data from the Department of Provincial Administration, Ministry of Interior, Thailand, in Excel format. Using Python, we extracted input features, including year, city, and total population, and subsequently organized this data into a MySQL database for further analysis [17].

The International Air Transport Association (IATA), a trade organization representing airlines worldwide, offers a variety of datasets pertaining to air travel and the airline industry. The IATA datasets cover a broad range of topics, including passenger traffic and revenue [14]. To analyze and identify the countries with the highest frequency of air travel to and from Thailand, we used data provided by the IATA. These statistics allowed us to gain insight into the most frequently traveled countries in terms of air transportation to and from Thailand.

FluNet is a global online platform and influenza surveillance system created and maintained by the National Influenza Centers (NICs) of the Global Influenza Surveillance and Response System (GISRS), as well as other national influenza reference laboratories that actively collaborate with GISRS. Data in FluNet is collected from these sources in addition to being uploaded from the WHO regional databases. This platform serves as an extensive database that facilitates the collection and distribution of influenza data from various countries

worldwide. FluNet provides information on influenza virus strains, epidemiological trends, and vaccine efficacy [9]. For our analysis and assessment of global influenza infection trends, we utilized influenza surveillance data obtained from FluNet.

## Data preprocessing

To standardize the data, we adopted a common population size of 100,000 as the denominator. This standardization facilitates meaningful comparisons across different regions or time periods. It adjusts for variations in population size, simplifying the assessment of the relative disease impact in various areas. Additionally, it reduces sensitivity to population size changes over time, which is crucial for tracking trends and making year-to-year comparisons. This approach is commonly utilized in numerous research studies [18,19]. In this study, we standardized the total number of cases for both Influenza and COVID-19 in each province. This standardization was achieved by converting the case counts into per 100,000 people for each province, using surveillance data from the Department of Disease Control, Ministry of Public Health and national population data from Department of Provincial Administration, Ministry of Interior in Thailand.

## Clustering time-series data with the DBSCAN algorithm

Density Based Spatial Clustering of Applications with Noise (DBSCAN) is an effective clustering algorithm that defines a cluster as the maximal set of density-connected points. It relies on the concepts of neighborhood and connectivity to determine clusters. This method utilizes two parameters: $\epsilon$ (epsilon) and MinPts. $\epsilon$ defines the maximum distance between two data points for them to be considered neighbors, effectively determining the size of the neighborhood around each data point. Meanwhile, the MinPts parameter sets the minimum number of data points necessary to establish a dense region. Data points that have at least MinPts neighbors within the $\epsilon$ distance are designated as core points.

DBSCAN classifies data points into three categories: core, border, and outlier (anomalous). Core points are defined as part of a dense region that have at least MinPts points within a specific distance of $\epsilon$. Border points are within the specified distance of a core point but do not have enough neighbors to be considered as a core point. Outlier points are neither core nor border points. They are typically isolated and not part of any cluster.

By distinguishing between these categories, DBSCAN effectively identifies dense regions as clusters while also capturing sparse regions and outliers. This flexibility makes DBSCAN a valuable algorithm for various clustering tasks. DBSCAN is known for its ability to discover clusters of varying shapes and sizes while effectively handling noise. However, selecting appropriate values for $\epsilon$ and MinPts can be a challenge, and the algorithm's performance may vary based on the dataset [20–22].

We configured the hyperparameters of the DBSCAN algorithm by setting MinPts to 2, ensuring it could capture any cluster with a membership size of 2 or more. Additionally, we conducted repeated experiments, varying the $\epsilon$ value from 0.1 to 99.9. Subsequently, we selected the smallest $\epsilon$ value that yielded the highest number of clusters, which were $\epsilon = 2$ for Pandemic Influenza, $\epsilon = 2.1$ for Endemic Influenza, and $\epsilon = 2.7$ for Pandemic COVID-19.

## Results

Utilizing DBSCAN clustering analysis, we have compiled the summarized findings of the clustering analysis for influenza and COVID-19, which are presented in Table 2. Additionally, a breakdown of provinces within each cluster for Pandemic Influenza, Endemic Influenza, and Pandemic COVID-19 is provided in Table 3–5. The analysis revealed noticeable variations in

**Table 2. Summary results from the clustering analysis of influenza and COVID-19 in Thailand.**

|  | Cluster -1 | | Cluster 0 | | Cluster 1 | | Cluster 2 | | Cluster 3 | |
| --- | --- | --- | --- | --- | --- | --- | --- | --- | --- | --- |
|  | Total | % | Total | % | Total | % | Total | % | Total | % |
| Pandemic Influenza | 61 | 77.92 | 11 | 14.29 | 2 | 2.60 | 3 | 3.90 | - | - |
| Endemic Influenza | 46 | 59.74 | 24 | 31.17 | 2 | 2.60 | 3 | 3.90 | 2 | 2.60 |
| Pandemic COVID-19 | 65 | 84.42 | 6 | 7.79 | 2 | 2.60 | 2 | 2.60 | 2 | 2.60 |

the spread patterns of influenza and COVID-19 among the majority of provinces in Thailand. The 77 provinces in Thailand were grouped into 4 to 5 clusters. The analysis revealed that 61 provinces for Pandemic Influenza and 65 provinces for Pandemic COVID-19 were clustered in Cluster -1, displayed distinct transmission patterns for respiratory diseases. In contrast, Endemic Influenza shown only 46 provinces in Cluster -1. In addition, Clusters 0, 1, 2, and 3 comprised provinces with comparable transmission patterns among their respective members. Pandemic Influenza had a total of 16 provinces in these clusters (11 in Cluster 0, 2 in Cluster 1, and 3 in Cluster 2), while Pandemic COVID-19 had 12 provinces in these clusters (6 in Cluster 0, 2 in Cluster 1, 2 in Cluster 2, and 2 in Cluster 3). In contrast, Endemic Influenza exhibited 31 provinces in these clusters (24 in Cluster 0, 2 in Cluster 1, 3 in Cluster 2, and 2 in Cluster 3). These results indicated that respiratory diseases shared similar spreading patterns during the initial stages of the global pandemic. As the pandemic influenza transitions to an endemic

**Table 3. Clustered provinces of Pandemic influenza in Thailand.**

| Cluster | Provinces |
| --- | --- |
| Cluster -1 | Bangkok, Bungkan, Buri Ram, Chachoengsao, Chai Nat, Chanthaburi, Chiang Mai, Chiang Rai, Chon Buri, Chumphon, Kamphaeng Phet, Kanchanaburi, Krabi, Lampang, Lamphun, Loei, Lop Buri, Mae Hong Son, Mukdahan, Nakhon Nayok, Nakhon Pathom, Nakhon Phanom, Nakhon Sawan, Nakhon Si Thammarat, Nan, Narathiwat, Nong Khai, Nonthaburi, P.Nakhon S.Ayutthaya, Pathum Thani, Phangnga, Phatthalung, Phayao, Phetchaburi, Phichit, Phitsanulok, Phrae, Phuket, Prachin Buri, Prachuap Khiri Khan, Ratchaburi, Rayong, Sakon Nakhon, Samut Prakan, Samut Sakhon, Samut Songkhram, Si Sa Ket, Sing Buri, Songkhla, Sukhothai, Suphan Buri, Surat Thani, Tak, Trang, Trat, Ubon Ratchathani, Udon Thani, Uthai Thani,Uttaradit, Yala, Yasothon |
| Cluster 0 | Amnat Charoen, Ang Thong, Chaiyaphum, Kalasin, Maha Sarakham, Nakhon Ratchasima, Nong Bua Lam Phu, Pattani, Roi Et, Sa Kaeo, Saraburi |
| Cluster 1 | Khon Kaen, Phetchabun |
| Cluster 2 | Ranong, Satun, Surin |

**Table 4. Clustered provinces of Endemic influenza in Thailand.**

| Cluster | Provinces |
| --- | --- |
| Cluster -1 | Ang Thong, Bangkok, Buri Ram, Chachoengsao, Chaiyaphum, Chanthaburi, Chiang Mai, Chiang Rai, Chon Buri, Chumphon, Kamphaeng Phet, Kanchanaburi, Lampang, Lamphun, Lop Buri, Mae Hong Son, Mukdahan, Nakhon Pathom, Nakhon Ratchasima, Nakhon Sawan, Nakhon Si Thammarat, Nan, Nong Khai, P.Nakhon S.Ayutthaya, Phangnga, Phatthalung, Phayao, Phetchaburi, Phitsanulok, Phrae, Phuket, Prachin Buri, Prachuap Khiri Khan, Ratchaburi, Rayong, Samut Prakan, Samut Sakhon, Samut Songkhram, Sukhothai, Surat Thani, Surin, Tak, Trang, Trat, Ubon Ratchathani, Uttaradit |
| Cluster 0 | Amnat Charoen, Bungkan, Chai Nat, Kalasin, Loei, Maha Sarakham, Nakhon Nayok, Nong Bua Lam Phu, Nonthaburi, Pathum Thani, Pattani, Phetchabun, Ranong, Roi Et, Sa Kaeo, Sakon Nakhon, Saraburi, Satun, Si Sa Ket, Sing Buri, Suphan Buri, Udon Thani, Uthai Thani, Yala |
| Cluster 1 | Khon Kaen, Phichit |
| Cluster 2 | Krabi, Narathiwat, Songkhla |
| Cluster 3 | Nakhon Phanom, Yasothon |

**Table 5. Clustered provinces of Pandemic COVID-19 in Thailand.**

| Cluster | Provinces |
|---|---|
| Cluster -1 | Ang Thong, Bangkok, Bungkan, Buri Ram, Chachoengsao, Chai Nat, Chanthaburi, Chiang Mai, Chiang Rai, Chon Buri, Chumphon, Kalasin, Kamphaeng Phet, Kanchanaburi, Khon Kaen, Krabi, Lampang, Lamphun, Loei, Lop Buri, Mae Hong Son, Nakhon Nayok, Nakhon Pathom, Nakhon Si Thammarat, Nan, Narathiwat, Nong Bua Lam Phu, Nong Khai, Nonthaburi, P.Nakhon S.Ayutthaya, Pathum Thani, Pattani, Phangnga, Phatthalung, Phayao, Phetchaburi, Phichit, Phrae, Phuket, Prachin Buri, Prachuap Khiri Khan, Ranong, Ratchaburi, Rayong, Sa Kaeo, Samut Prakan, Samut Sakhon, Samut Songkhram, Saraburi, Satun, Si Sa Ket, Sing Buri, Songkhla, Sukhothai, Suphan Buri, Surat Thani, Surin, Tak, Trang, Trat, Ubon Ratchathani, Udon Thani, Uthai Thani, Uttaradit, Yala |
| Cluster 0 | Amnat Charoen, Chaiyaphum, Mukdahan, Nakhon Phanom, Sakon Nakhon, Yasothon |
| Cluster 1 | Maha Sarakham, Roi Et |
| Cluster 2 | Nakhon Ratchasima, Phetchabun |
| Cluster 3 | Nakhon Sawan, Phitsanulok |

phase and populations receive vaccinations, the domestic spreading pattern of influenza changes. These changes in characteristic transmission trends can be influenced by various factors, including shifts in population immunity due to vaccination strategies, seasonal variations, improvements in healthcare preparedness encompassing surveillance and early detection, alterations in travel patterns, and the evolution of the virus. It's important that the transition from pandemic to endemic is a complex process influenced by various factors, and the specific changes in transmission patterns can vary by region and over time. Public health authorities should maintain monitoring of these patterns closely to adapt control measures and vaccination strategies accordingly.

Furthermore, to enhance the analytical perspective and improve the understanding of the results obtained from the clustering analysis, we performed visualization of the clustering analysis outcomes, particularly focusing on the geographical aspects of the regions. Clustering maps illustrating the clusters of influenza and COVID-19 cases are shown in Figs 2–4. These visual representations offer valuable insights into the spatial distribution and clustering patterns of influenza and COVID-19 cases, aiding in a more comprehensive analysis of the data.

The provinces represented by the members of Cluster -1, as shown in Figs 2–4, are characterized by a lack of significant relations with other provinces. These provinces are primarily located in the northern, central, southern, and eastern regions of Thailand. In contrast, members of Clusters 0, 1, 2, and 3 are predominantly located in the northeastern region.

These visualizations highlight several noteworthy observations. Provinces in the northern region consistently exhibit unique transmission patterns across all experiments. In addition, during both Pandemic Influenza and Pandemic COVID-19, nearly every province in the southern region except three provinces during Pandemic influenza displays a unique transmission pattern. However, in the case of Endemic Influenza, seven provinces have been incorporated into clusters. Similarly, in the northeast, western, and central regions, several provinces have been integrated into clusters during the Endemic Influenza phase. These provinces represent the transition from a pandemic to an endemic state, wherein transmission patterns become increasingly reliant on domestic factors, resulting in similarities with their cluster members.

These findings demonstrate a similarity in the domestic spreading patterns of influenza and COVID-19 during the initial stages of the global pandemic. Not only do the results reveal similarities in terms of the number of cluster members but also in the spatial distribution patterns. This suggests that, during the initial periods of the global pandemic, the two respiratory diseases spread throughout the country similarly.

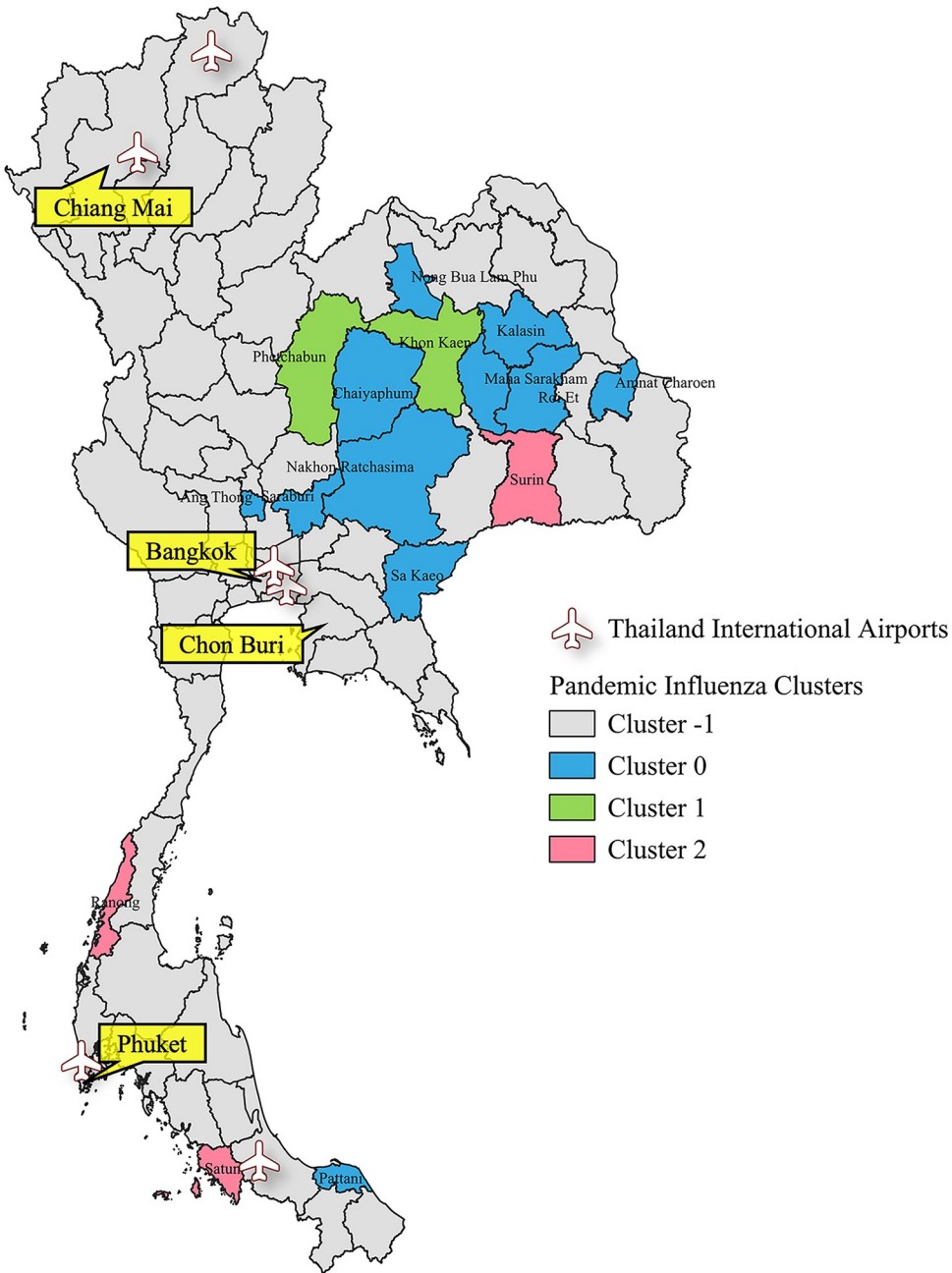

**Fig 2. Visualization of clustering analysis results for Pandemic Influenza in Thailand.** The map in this figure was created with QGIS version 3.30.2. Source of shapefile: https://www.borntodev.com/2019/07/01/ under a CC BY license, with permission from BorntoDev Co., Ltd., original copyright 2022.

Additionally, our analysis revealed that Roi Et province and Maha Sarakham province consistently belonged to the same cluster in all clustering analyses conducted. These results indicate a similar pattern in infection trends between these two provinces, distinguishing them from other provinces, particularly in the case of Pandemic Influenza and Pandemic COVID-19. To enhance the visualization, we plotted the trends of influenza and COVID-19, as illustrated in Figs 5–7. These visual representations provide a clearer depiction of patterns and changes in the occurrence of these respiratory diseases over a specified period.

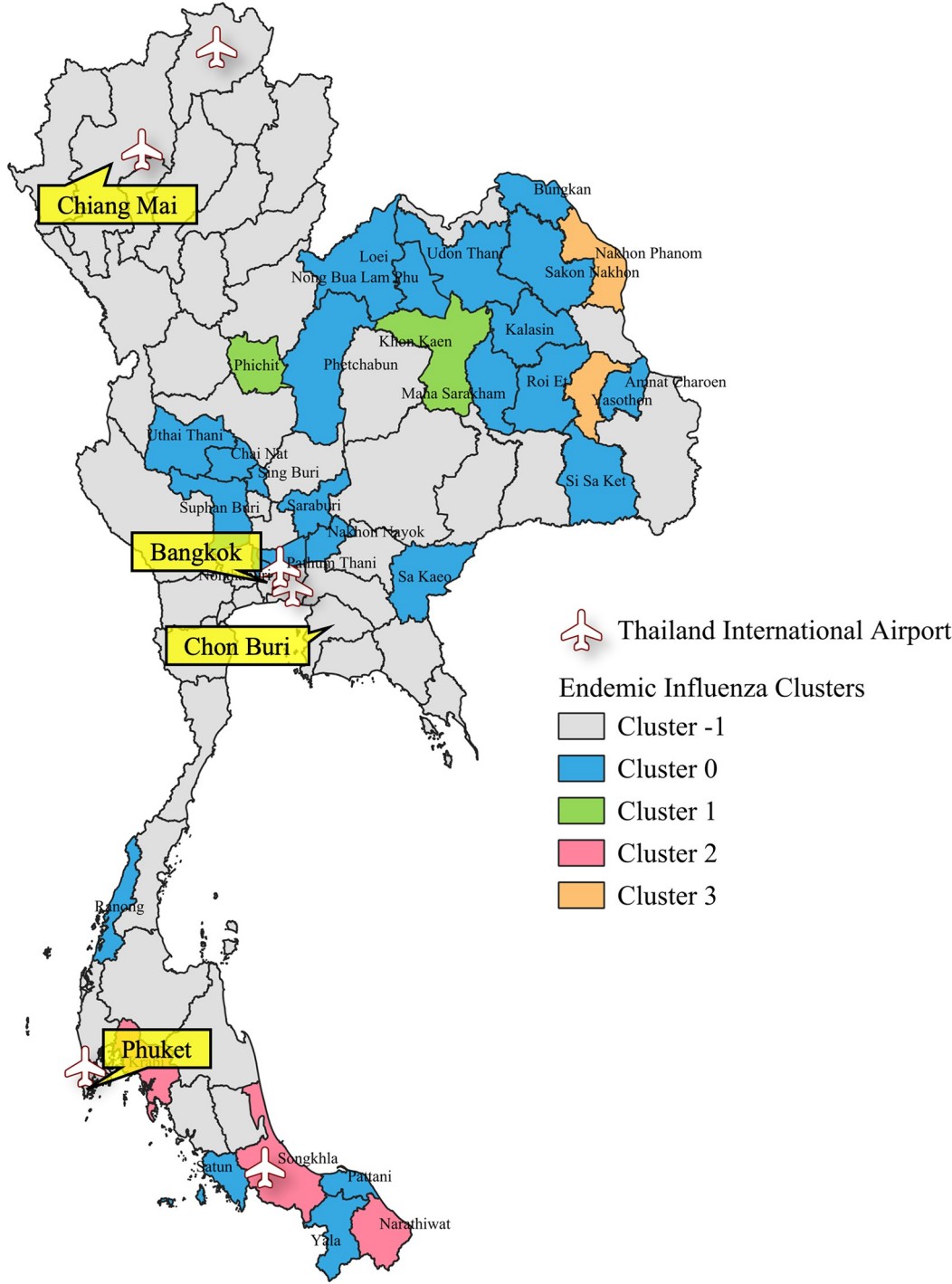

**Fig 3. Visualization of clustering analysis results for Endemic Influenza in Thailand.** The map in this figure was created with QGIS version 3.30.2. Source of shapefile: https://www.borntodev.com/2019/07/01/ under a CC BY license, with permission from BorntoDev Co., Ltd., original copyright 2022.

To compare the trends of Pandemic Influenza, Endemic Influenza, and Pandemic COVID-19, we visualized the monthly infection patterns in Roi Et and Maha Sarakham provinces, alongside other significant provinces in Thailand. These major provinces include Bangkok, the

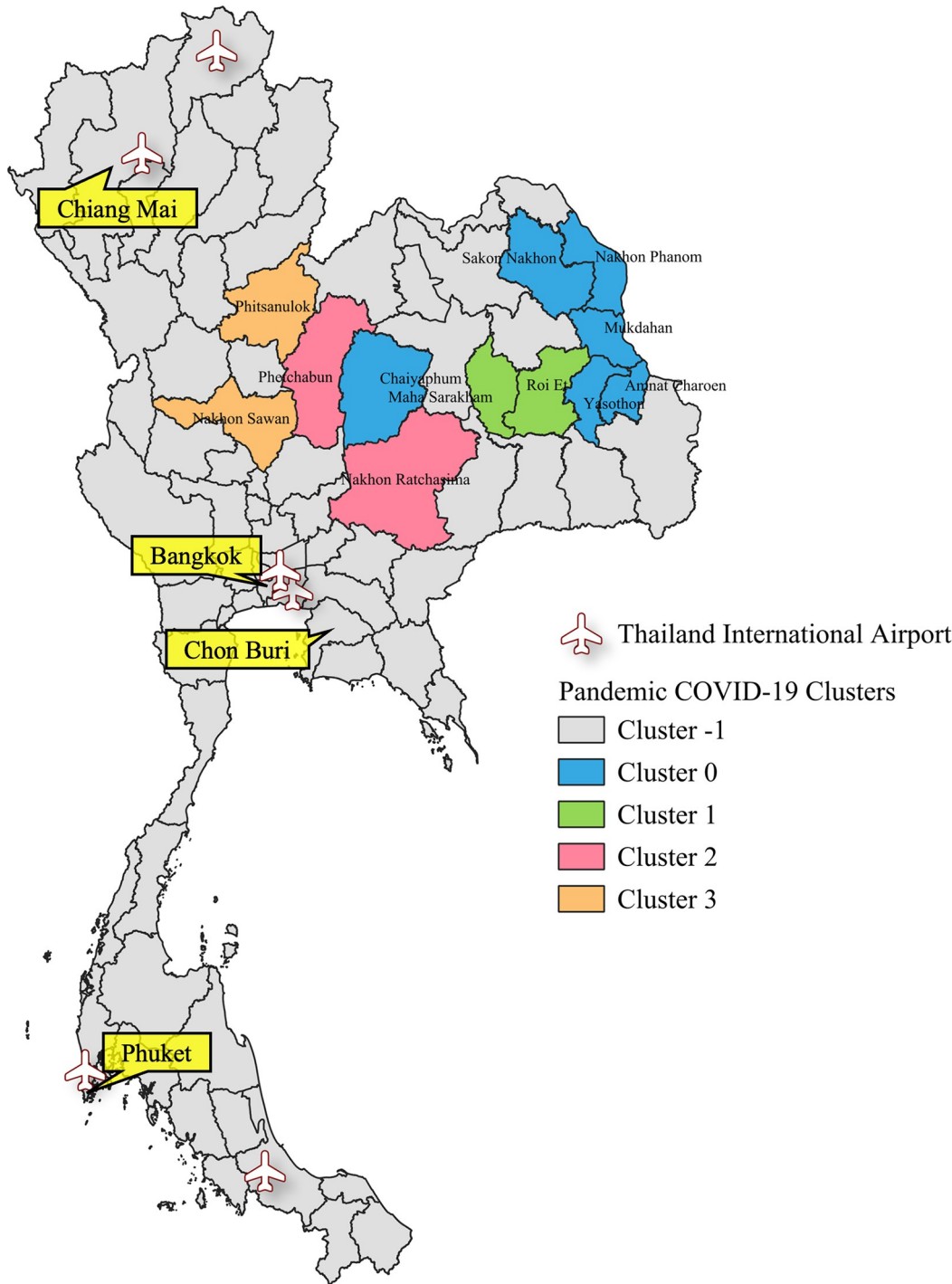

**Fig 4. Visualization of clustering analysis results for Pandemic COVID-19 in Thailand.** The map in this figure was created with QGIS version 3.30.2. Source of shapefile: https://www.borntodev.com/2019/07/01/ under a CC BY license, with permission from BorntoDev Co., Ltd., original copyright 2022.

capital city of Thailand; Chiang Mai province, a popular tourist destination in the northern region; Chon Buri province, a tourist hotspot in the eastern region; and Phuket province, a renowned tourist attraction in the southern region. The findings indicated that, in comparison

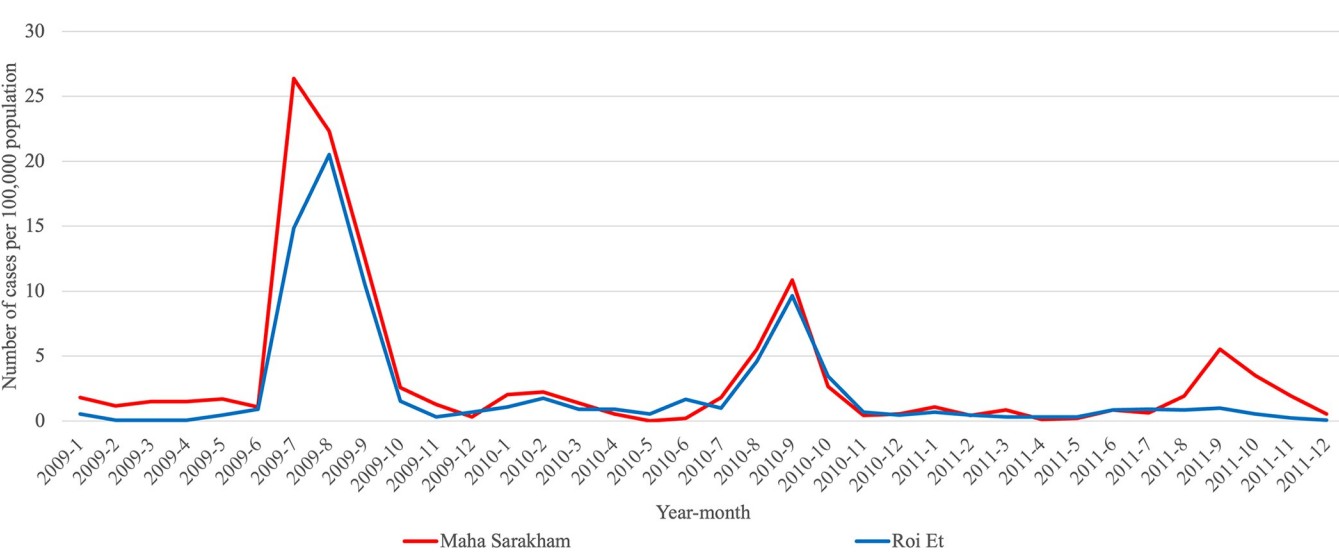

**Fig 5. Monthly trend of influenza calculated per 100,000 people in Roi Et province and Maha Sarakham province during 2009–2011.**

to the major provinces, Roi Et province and Maha Sarakham province exhibited a lower infection rate. Interestingly, these two provinces displayed remarkably similar trends to each other. Conversely, the trends in Roi Et province and Maha Sarakham province differed significantly from those observed in other major provinces, as shown in Figs 8–10. This divergence suggests the presence of unique factors that warrant further investigation, making it an interesting case for future study.

Identifying the key factors responsible for the similarity in cluster membership, as well as the differences from other clusters, could prove invaluable for effective public health management. It can assist in the development of strategies to control disease spread and mitigate its impact on public health.

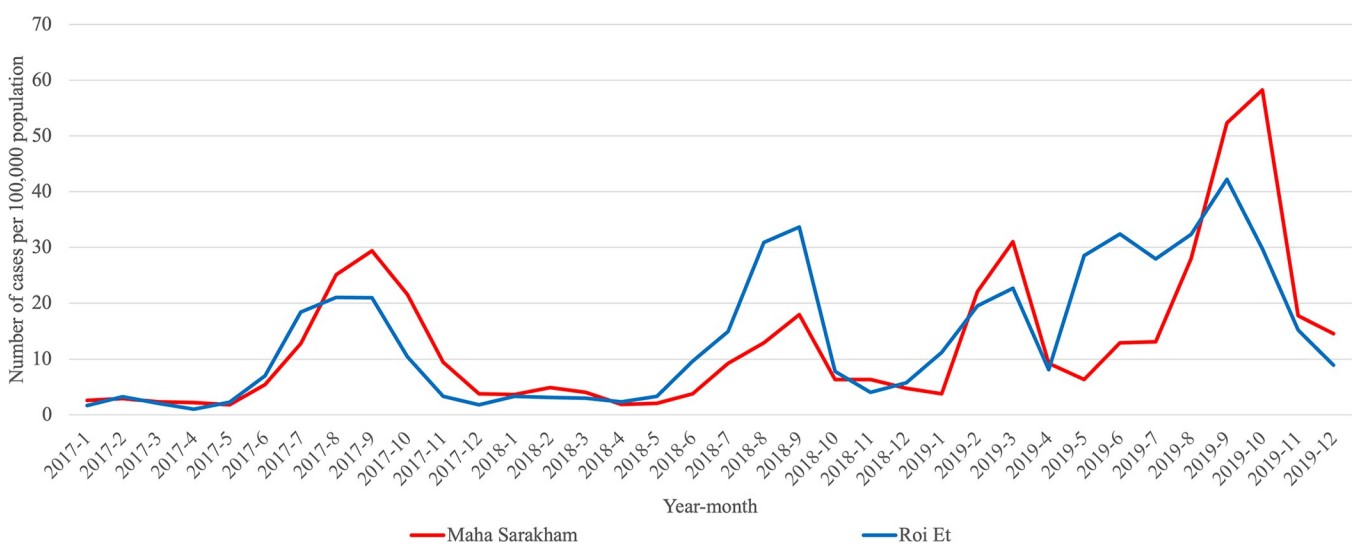

**Fig 6. Monthly trend of influenza calculated per 100,000 people in Roi Et province and Maha Sarakham province during 2017–2019.**

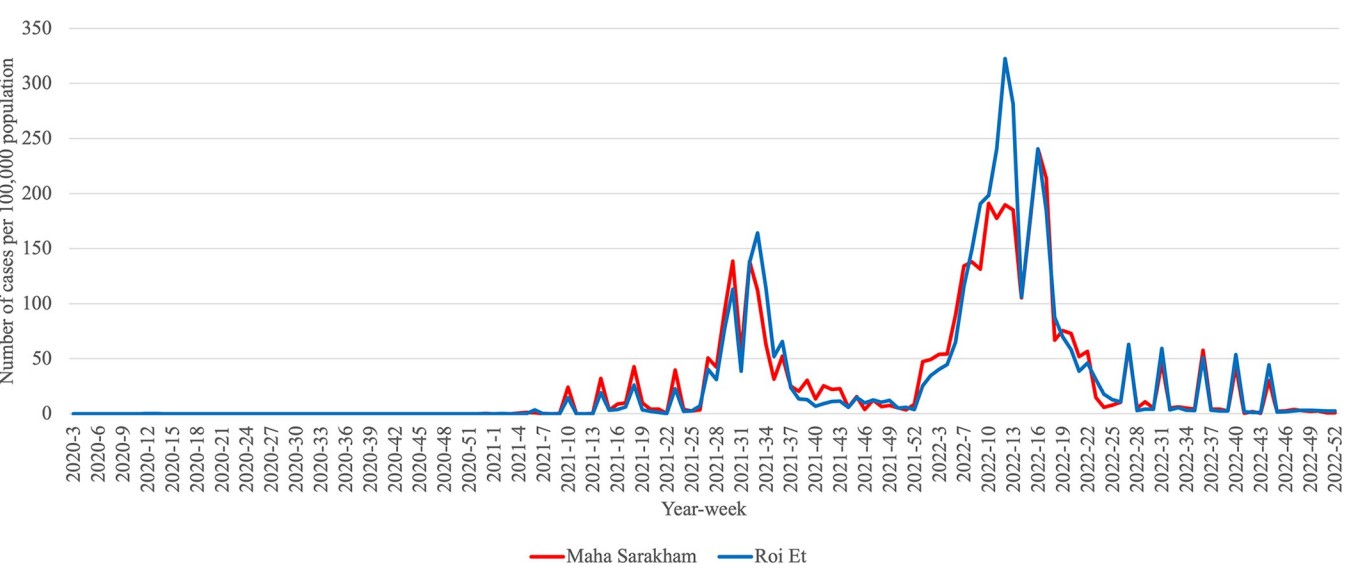

**Fig 7. Weekly trend of COVID-19 calculated per 100,000 people in Roi Et province and Maha Sarakham province during 2020–2022.**

## Conclusions

In this study, we employed DBSCAN clustering analysis to explore the transmission patterns of influenza and COVID-19 in Thailand. The analysis revealed similarities in the total number of cluster members and spatial distribution between Pandemic Influenza and Pandemic COVID-19, indicating comparable spreading patterns during the early stages of global pandemics. However, Endemic Influenza exhibited distinct patterns, suggesting changes in transmission dynamics as influenza transitions to an endemic phase and populations are vaccinated. This suggests that the spread pattern of COVID-19 in Thailand may resemble that of Endemic Influenza following the transition from pandemic to endemic status. The

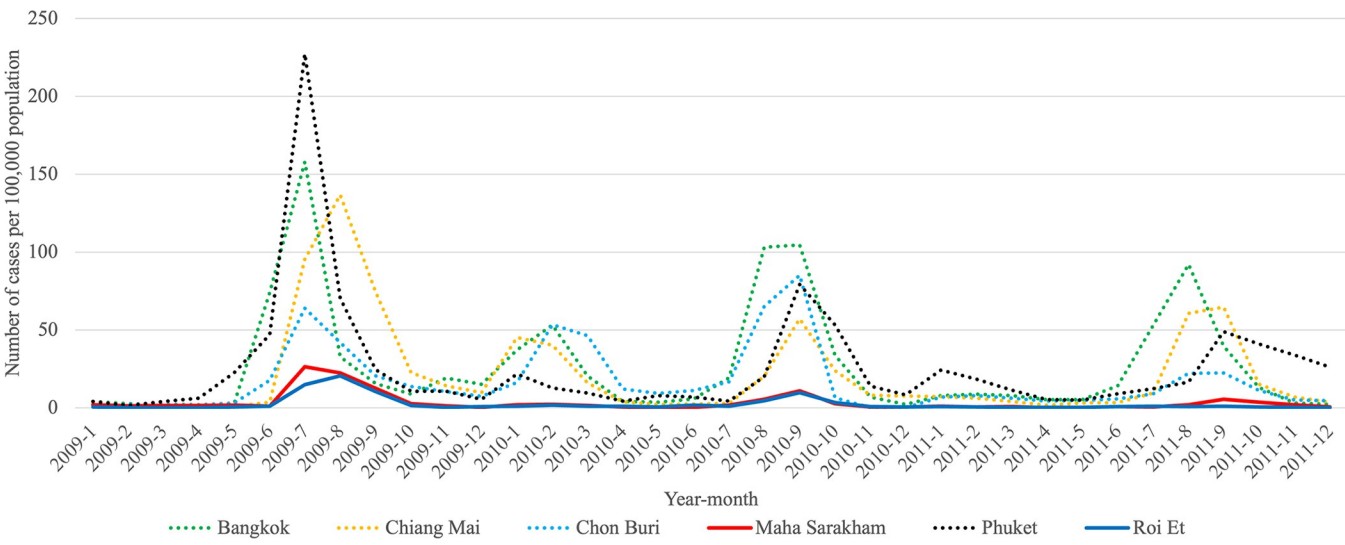

**Fig 8. Monthly trend of influenza calculated per 100,000 people in Roi Et province and Maha Sarakham province with major provinces during 2009–2011.**

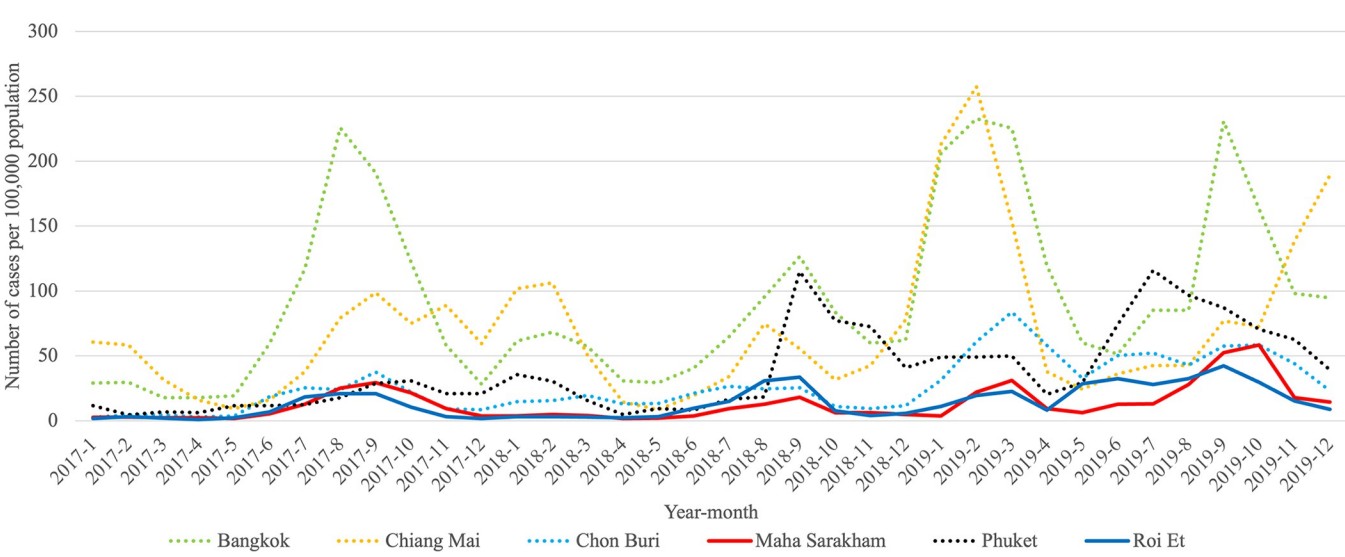

**Fig 9. Monthly trend of influenza calculated per 100,000 people in Roi Et province and Maha Sarakham province with major provinces during 2017–2019.**

discovery of a strongly similar trend between Roi Et province and Maha Sarakham province is noteworthy. This implies the existence of common factors, such as weather conditions, human transmission, and local festivals, which impact the spread of respiratory diseases in these provinces. Further investigations and experiments on this topic may yield valuable insights for future research. Additionally, distinctive patterns in the spatial distribution of Pandemic Influenza and Pandemic COVID-19 clusters were observed in the northern, southern, eastern, and central regions, as opposed to other areas. This can be attributed to factors such as population density, human crowding, rapid viral evolution, environmental influences, and international travel patterns. These regions are geographically situated in proximity to international airports in Thailand and serve as popular tourist destinations or the country's capital that experience

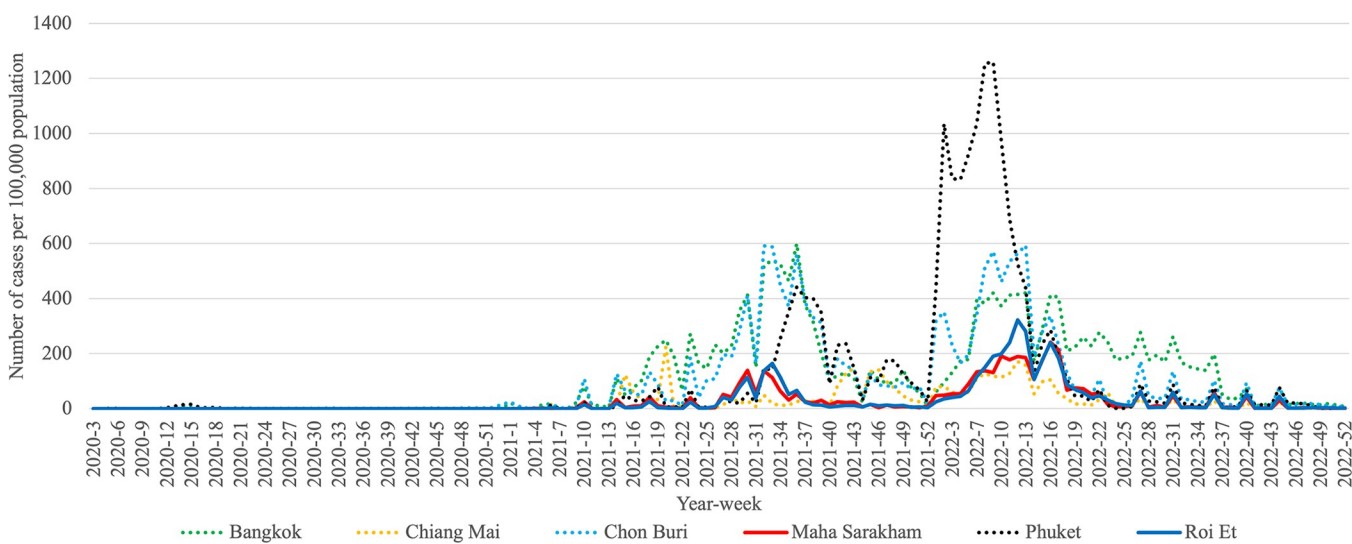

**Fig 10. Weekly trend of COVID-19 calculated per 100,000 people in Roi Et province and Maha Sarakham province with major provinces during 2020–2022.**

high levels of human crowding and international travel. These factors contribute to the changing transmission patterns of various diseases. Furthermore, the government's response to disease outbreaks, including measures such as lockdown restrictions, testing policies, contact tracing, and vaccination efforts, also influences spreading patterns. We hope that our study will provide valuable information to assist governments in formulating policies against future pandemics and navigating the transition to an endemic phase. This includes identifying the priorities for different regions based on their unique characteristics and responses. Future studies will incorporate data on government policies and domestic travel tracking to enhance our analysis.

## Author Contributions

**Conceptualization:** Insung Ahn.

**Data curation:** Thanin Methiyothin.

**Formal analysis:** Thanin Methiyothin.

**Funding acquisition:** Insung Ahn.

**Investigation:** Thanin Methiyothin, Insung Ahn.

**Methodology:** Thanin Methiyothin.

**Project administration:** Insung Ahn.

**Resources:** Insung Ahn.

**Software:** Thanin Methiyothin.

**Supervision:** Insung Ahn.

**Validation:** Thanin Methiyothin.

**Visualization:** Thanin Methiyothin.

**Writing – original draft:** Thanin Methiyothin.

**Writing – review & editing:** Insung Ahn.

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
