## [Decision Letter · Decision Letter 0]

14 Aug 2023

PONE-D-23-19116Comparison of Geological Clusters between Influenza and COVID-19 in Thailand with Unsupervised Clustering AnalysisPLOS ONE

Dear Dr. Ahn,

Thank you for submitting your manuscript to PLOS ONE. After careful consideration, we feel that it has merit but does not fully meet PLOS ONE’s publication criteria as it currently stands. Therefore, we invite you to submit a revised version of the manuscript that addresses the points raised during the review process. Please submit your revised manuscript by Sep 28 2023 11:59PM. If you will need more time than this to complete your revisions, please reply to this message or contact the journal office at plosone@plos.org. Please include the following items when submitting your revised manuscript:A rebuttal letter that responds to each point raised by the academic editor and reviewer(s). You should upload this letter as a separate file labeled 'Response to Reviewers'.A marked-up copy of your manuscript that highlights changes made to the original version. You should upload this as a separate file labeled 'Revised Manuscript with Track Changes'.An unmarked version of your revised paper without tracked changes. You should upload this as a separate file labeled 'Manuscript'.

We look forward to receiving your revised manuscript.

Kind regards,

Samrat Kumar Dey, MSc. Engg.

Academic Editor

PLOS ONE

3. We note that Figures 2,3 and 4 in your submission contain [map/satellite] images which may be copyrighted. All PLOS content is published under the Creative Commons Attribution License (CC BY 4.0), which means that the manuscript, images, and Supporting Information files will be freely available online, and any third party is permitted to access, download, copy, distribute, and use these materials in any way, even commercially, with proper attribution. For these reasons, we cannot publish previously copyrighted maps or satellite images created using proprietary data, such as Google software (Google Maps, Street View, and Earth). For more information, see our copyright guidelines: http://journals.plos.org/plosone/s/licenses-and-copyright.

a. You may seek permission from the original copyright holder of Figures 2,3 and 4 to publish the content specifically under the CC BY 4.0 license. 

Reviewers' comments:

Reviewer's Responses to Questions

**Comments to the Author**

1. Is the manuscript technically sound, and do the data support the conclusions?

Reviewer #1: Partly

Reviewer #2: Partly

2. Has the statistical analysis been performed appropriately and rigorously? 

Reviewer #1: N/A

Reviewer #2: Yes

3. Have the authors made all data underlying the findings in their manuscript fully available?

Reviewer #1: Yes

Reviewer #2: Yes

4. Is the manuscript presented in an intelligible fashion and written in standard English?

Reviewer #1: Yes

Reviewer #2: Yes

5. Review Comments to the Author

Reviewer #1: Revision of the article «Comparison of Geological Clusters between Influenza and COVID-19 in Thailand with Unsupervised Clustering Analysis»

The article aimed to investigate the transmission patterns of influenza and COVID-19 in Thailand using unsupervised clustering analysis. The findings of this study have important implications for public health management and preparedness for future pandemics or endemic phases. I have outlined my comments and suggestions below to improve the manuscript for potential publication:

General Comments:

1. Abstract:

The abstract provides a concise summary of the study. However, it would be beneficial to include specific findings or key results from the analysis in the abstract to make it more informative.

2. Introduction:

The introduction provides a good background of the study, but it can be further improved by including more recent references related to COVID-19 and influenza, especially since the pandemic has likely progressed since the time of your knowledge cutoff. Additionally, please ensure that all references cited in the introduction are appropriately cited in the main text.

3. Methods:

Provide more details on the data collection process from the Department of Disease Control (DDC), including any potential limitations or biases in the datasets used.

Clarify the rationale for averaging the influenza surveillance data into a daily time-series format before converting it into a weekly time-series format.

Explain how you determined the optimal values for ε (epsilon) and MinPts parameters for the DBSCAN algorithm.

4. Results:

Ensure that all tables and figures are properly referenced in the main text. Additionally, consider providing more descriptive captions for figures to enhance their understanding.

Provide more detailed insights and interpretations of the clustering analysis results. Discuss the significance of the observed variations in the spread patterns of influenza and COVID-19 among different provinces in Thailand.

5. Discussion:

It is crucial to include a dedicated discussion section to interpret the study's findings in light of the existing literature. Discuss the implications of the similarities and differences in the clustering patterns of influenza and COVID-19 during the pandemic and endemic phases. Relate your findings to the broader context of public health management and preparedness for future pandemics.

6. Language and Clarity:

Proofread the manuscript thoroughly for language errors and clarity. Consider revising some sentences to improve readability.

Specific Comments

Abstact:

While the objective is implied, it could be explicitly stated at the beginning of the abstract. For example: "The objective of this study was to identify and compare the transmission patterns of influenza and COVID-19 in Thailand."

Provide more specific details about the findings. Instead of using terms like "similarities" and "different patterns," mention the specific aspects that were similar and different between influenza and COVID-19 in the pandemic and endemic stages.

Include a sentence at the end of the abstract that highlights the significance of the study's findings and how they can contribute to public health management.

Introduction

1.Begin the introduction by providing a concise background on the worldwide outbreak of the coronavirus disease (COVID-19) pandemic in early 2020 and its significant impact on public health. Include statistics or references to highlight the scale of the pandemic and its global consequences.

2.Emphasize the importance of understanding the transmission patterns of respiratory pathogens, such as influenza, in the context of the COVID-19 pandemic. Briefly discuss the potential implications of changes in transmission patterns for public health management and preparedness for future pandemics.

3.Provide a more detailed and chronological account of the 2009 influenza pandemic (H1N1) to establish its significance as the first human influenza pandemic in the 21st century. Include the number of countries affected, confirmed cases, and fatalities to underscore its impact on global health.

4.Add more information on the public health measures implemented during the COVID-19 pandemic and their impact on mitigating the spread of both influenza and COVID-19. Include references to studies or official reports that support the claim of lower influenza cases and associated hospitalizations during the pandemic.

5.Clarify the connection between COVID-19 and influenza as both being respiratory infections that initially lacked vaccination-based immunity in the population.

6.While the general objective of identifying spread patterns in specific regions and provinces is mentioned, it could be more explicitly stated at the end of the introduction.

7.Clearly state that your research aims to provide quantitative and spatial information to aid public health management in preparing for future pandemics or transitioning into an endemic phase. Highlight the practical implications of your findings for public health strategies and decision-making.

Example of reference you may used in the text:

Kumar, S., Havens, P., Chusid, M., Willoughby, R., Simpson, P., & Henrickson, K. (2010). Clinical and Epidemiologic Characteristics of Children Hospitalized With 2009 Pandemic H1N1 Influenza A Infection. The Pediatric Infectious Disease Journal, 29, 591-594. https://doi.org/10.1097/INF.0b013e3181d73e32.

Yousfi, N., Bragazzi, N. L., Briki, W., Zmijewski, P., & Chamari, K. (2020). The COVID-19 pandemic: how to maintain a healthy immune system during the lockdown–a multidisciplinary approach with special focus on athletes. Biology of sport, 37(3), 211-216.

González-Sanguino, C., Ausín, B., Castellanos, M., Sáiz, J., López-Gómez, A., Ugidos, C., & Muñoz, M. (2020). Mental health consequences during the initial stage of the 2020 Coronavirus pandemic (COVID-19) in Spain. Brain, Behavior, and Immunity, 87, 172 - 176. https://doi.org/10.1016/j.bbi.2020.05.040.

Soo, R., Chiew, C., Ma, S., Pung, R., & Lee, V. (2020). Decreased Influenza Incidence under COVID-19 Control Measures, Singapore. Emerging Infectious Diseases, 26, 1933 - 1935. https://doi.org/10.3201/eid2608.201229.

Arellanos-Soto, D., Padilla-Rivas, G., Ramos-Jiménez, J., Galán-Huerta, K., Lozano-Sepúlveda, S., Martínez-Acuña, N., Treviño-Garza, C., Montes-de-Oca-Luna, R., de-la-O-Cavazos, M., & Rivas-Estilla, A. (2021). Decline in influenza cases in Mexico after the implementation of public health measures for COVID-19. Scientific Reports, 11. https://doi.org/10.1038/s41598-021-90329-w.

Methods

1.Provide more details about the data sources from the DDC, such as the methods used for data collection, the geographic coverage (i.e., data from each province), and any potential limitations or biases in the data that should be considered in the analysis.

2.In the section about the International Air Transport Association (IATA) datasets, specify the time period covered by the air travel data used to identify the countries with the highest frequency of air travel to and from Thailand. Explain how this data will be integrated into the analysis of transmission patterns.

3.When converting the influenza surveillance data from monthly to daily and then to weekly time-series formats, provide a clear rationale for choosing these time intervals. Explain how this data transformation process affects the subsequent analysis.

4.Describe the normalization process used to convert the total number of influenza cases for each province into the number of cases per 100,000 people. Clarify why this normalization method was chosen and how it enables meaningful comparisons between provinces with different population sizes.

5.Provide a more detailed explanation of the Density Based Spatial Clustering of Applications with Noise (DBSCAN) algorithm. Explain the concepts of neighborhood and connectivity used to determine clusters and how the parameters (ε and MinPts) were selected or optimized for the for the DBSCAN algorithm.

6.Clarify how the three categories of data points (core, border, and outlier) are identified and utilized in the DBSCAN clustering analysis. Elaborate on how these categories contribute to identifying dense regions and outliers in the data.

7.Mention any potential limitations or challenges associated with using the DBSCAN algorithm for clustering time-series data. Discuss how these challenges were addressed to ensure the reliability and validity of the clustering results.

Results

1.Clearly state the main findings of the clustering analysis in a concise manner at the beginning of the "Results" section.

2.Provide a summary of the number of clusters identified and the provinces included in each cluster for both influenza and COVID-19 datasets.

3.When discussing the results in the text, highlight the most significant observations from Table 2. Emphasize the variations in transmission patterns between the different stages of influenza (pandemic vs. endemic) and COVID-19, and explain the implications of these differences on public health management and future pandemic preparedness.

4.Provide a more detailed interpretation of the clustering maps shown in Figs 2-4. Describe the geographic distribution of clusters in Thailand, and discuss any regional patterns or trends observed.

5.Provide a clear explanation of the fluctuations or trends observed in the infection rates of influenza and COVID-19 in Roi Et and Maha Sarakham provinces.

6.Discuss the practical implications of the observed differences in infection trends among the major provinces. Address how these findings can inform targeted public health interventions and resource allocation in different regions of Thailand

Reviewer #2: This study represents the use of unsupervised clustering techniques (especially, DBSSCAN) to identify clusters of disease trends in Thailand. Three distinct surveillance datasets, namely the pandemic influenza outbreak, influenza in the endemic stage, and the early stages of COVID-19, were then used as time-series data of the number of infected people in those periods. They revealed similarities between influenza and COVID-19 transmission patterns during the pandemic phases, including the number of provinces in the clusters and spatial distribution patterns. A pair of provinces with highly similar spreading patterns during both the pandemic stages of influenza and COVID-19 was also reported. It was a great analysis to study the patterns of the endemic in Thailand. However, there are some concerns and limitations that need addressing:

1. The authors should explore the statistics of the data used and perform necessary preprocessing before applying clustering.

2. The choice of weekly data for analysis needs clarification, as it might result in repetitive values, making it similar to using monthly data. Why would the authors perform the analysis for weekly data?

3. The time-series data for all provinces should be visually presented to observe any common distribution patterns among them. The plot of these can show us the outliers and common groups of provinces. These can also be applied to all three data sets.

4. The authors should explain how they selected the number of clusters in the DBSCAN algorithm and explore different parameters to improve clustering results. The reasons for choosing DBSCAN as the clustering algorithm and determining optimal parameters (e.g., epsilon and minimum points) should be elaborated.

5. The authors should address the situations or policies that could have contributed to the resulting clusters, such as the behavior of people, travel patterns, or government policies. Practical implications for policymakers and public health officials need to be discussed, explaining how this knowledge can inform decision-making and response strategies during future pandemics or endemic phases. What are the situations or policies behind the resulting clusters? These should be clearly addressed. It might be because of the behavior of the Thai people or because workers always come to Bangkok and go back to their homes during long holidays or a knock-down policy.

6. The authors should compare their findings with previous studies on similar topics to validate their results and strengthen their conclusions. Some other clustering approaches may provide different clustering results.

7. Limitations, including inherent biases in clustering analysis and potential missed aspects of disease transmission, should be acknowledged.

8. All figures should be properly labeled with axis titles, and higher-resolution figures should be provided to improve clarity. It may be not necessary to have Figure 1 in the circumstance of the analysis. Only report the number in the text would be enough.

9. There is a lack of discussion part that the author can discuss the methods, results, practical implications, and limitation of their study.

6. PLOS authors have the option to publish the peer review history of their article (what does this mean?). If published, this will include your full peer review and any attached files.

Reviewer #1: No

Reviewer #2: No

---

## [Author Response · Author response to Decision Letter 0]

28 Nov 2023

Thank you very much to reviewers for your time, expertise, and valuable feedback. Your contributions have greatly enhanced the quality of our manuscript. I've provided a summary of my responses below.

Reviewer 1

Specific Comments 

Abstact:

While the objective is implied, it could be explicitly stated at the beginning of the abstract. For example: "The objective of this study was to identify and compare the transmission patterns of influenza and COVID-19 in Thailand."

 Provide more specific details about the findings. Instead of using terms like "similarities" and "different patterns," mention the specific aspects that were similar and different between influenza and COVID-19 in the pandemic and endemic stages.

Include a sentence at the end of the abstract that highlights the significance of the study's findings and how they can contribute to public health management.

Introduction

1. Begin the introduction by providing a concise background on the worldwide outbreak of the coronavirus disease (COVID-19) pandemic in early 2020 and its significant impact on public health. Include statistics or references to highlight the scale of the pandemic and its global consequences.

o I’ve added background information in line 49-53.

2. Emphasize the importance of understanding the transmission patterns of respiratory pathogens, such as influenza, in the context of the COVID-19 pandemic. Briefly discuss the potential implications of changes in transmission patterns for public health management and preparedness for future pandemics. 

o I’ve made additions on lines 68-81.

3. Provide a more detailed and chronological account of the 2009 influenza pandemic (H1N1) to establish its significance as the first human influenza pandemic in the 21st century. Include the number of countries affected, confirmed cases, and fatalities to underscore its impact on global health.

o I’ve added background information in line 57-64

4. Add more information on the public health measures implemented during the COVID-19 pandemic and their impact on mitigating the spread of both influenza and COVID-19. Include references to studies or official reports that support the claim of lower influenza cases and associated hospitalizations during the pandemic. 

o I’ve made additions on lines 86-92.

5. Clarify the connection between COVID-19 and influenza as both being respiratory infections that initially lacked vaccination-based immunity in the population. 

o I’ve added references in line 68.

6. While the general objective of identifying spread patterns in specific regions and provinces is mentioned, it could be more explicitly stated at the end of the introduction. 

o I’ve made additions on lines 141-145.

7. Clearly state that your research aims to provide quantitative and spatial information to aid public health management in preparing for future pandemics or transitioning into an endemic phase. Highlight the practical implications of your findings for public health strategies and decision-making. 

o I’ve made additions on lines 141-145.

Methods

1. Provide more details about the data sources from the DDC, such as the methods used for data collection, the geographic coverage (i.e., data from each province), and any potential limitations or biases in the data that should be considered in the analysis. 

o I’ve made additions on lines 167-169 and 172-183.

2. In the section about the International Air Transport Association (IATA) datasets, specify the time period covered by the air travel data used to identify the countries with the highest frequency of air travel to and from Thailand. Explain how this data will be integrated into the analysis of transmission patterns. 

o This data serves as supporting information in the introduction section, where it is discussed that the first COVID-19 infection outside China and the first case in Thailand were in line with the summary of air travel passenger data. This data revealed that passengers from China ranked first and had significantly higher numbers compared to passengers from other regions.

3. When converting the influenza surveillance data from monthly to daily and then to weekly time-series formats, provide a clear rationale for choosing these time intervals. Explain how this data transformation process affects the subsequent analysis. 

o I have reverted to using the original monthly data to prevent minor errors. The results do not show significant deviations from the previous experiment. I have already incorporated these revisions into the updated results.

4. Describe the normalization process used to convert the total number of influenza cases for each province into the number of cases per 100,000 people. Clarify why this normalization method was chosen and how it enables meaningful comparisons between provinces with different population sizes. 

o I’ve made additions on lines 201-212.

5. Provide a more detailed explanation of the Density Based Spatial Clustering of Applications with Noise (DBSCAN) algorithm. Explain the concepts of neighborhood and connectivity used to determine clusters and how the parameters (ε and MinPts) were selected or optimized for the for the DBSCAN algorithm. 

o I’ve made modifications and additions on lines 218-227 and 251-255.

6. Clarify how the three categories of data points (core, border, and outlier) are identified and utilized in the DBSCAN clustering analysis. Elaborate on how these categories contribute to identifying dense regions and outliers in the data.

o I’ve made modifications and additions on lines 218-227.

7. Mention any potential limitations or challenges associated with using the DBSCAN algorithm for clustering time-series data. Discuss how these challenges were addressed to ensure the reliability and validity of the clustering results.

o I’ve made additions on lines 247-250.

 Results

1. Clearly state the main findings of the clustering analysis in a concise manner at the beginning of the "Results" section. 

o I've made modifications and additions on lines 258-274.

2. Provide a summary of the number of clusters identified and the provinces included in each cluster for both influenza and COVID-19 datasets.

o I’ve made additions on tables 3-5.

3. When discussing the results in the text, highlight the most significant observations from Table 2. Emphasize the variations in transmission patterns between the different stages of influenza (pandemic vs. endemic) and COVID-19, and explain the implications of these differences on public health management and future pandemic preparedness.

o I’ve made additions on lines 277-284.

4. Provide a more detailed interpretation of the clustering maps shown in Figs 2-4. Describe the geographic distribution of clusters in Thailand, and discuss any regional patterns or trends observed.

o I’ve made additions on lines 339-347.

5. Provide a clear explanation of the fluctuations or trends observed in the infection rates of influenza and COVID-19 in Roi Et and Maha Sarakham provinces.

o I’ve made additions on lines 384-30.

6. Discuss the practical implications of the observed differences in infection trends among the major provinces. Address how these findings can inform targeted public health interventions and resource allocation in different regions of Thailand 

o I’ve made additions on lines 408-411.

Reviewer2

Reviewer #2: This study represents the use of unsupervised clustering techniques (especially, DBSSCAN) to identify clusters of disease trends in Thailand. Three distinct surveillance datasets, namely the pandemic influenza outbreak, influenza in the endemic stage, and the early stages of COVID-19, were then used as time-series data of the number of infected people in those periods. They revealed similarities between influenza and COVID-19 transmission patterns during the pandemic phases, including the number of provinces in the clusters and spatial distribution patterns. A pair of provinces with highly similar spreading patterns during both the pandemic stages of influenza and COVID-19 was also reported. It was a great analysis to study the patterns of the endemic in Thailand. However, there are some concerns and limitations that need addressing:

1. The authors should explore the statistics of the data used and perform necessary preprocessing before applying clustering. 

o I decided to revert to use original monthly data and added explanation about preprocessing data in line 167-169, 172-174, 180-183, and 201-212.

2. The choice of weekly data for analysis needs clarification, as it might result in repetitive values, making it similar to using monthly data. Why would the authors perform the analysis for weekly data? 

o I have reverted to using the original monthly data to prevent minor errors. The results do not show significant deviations from the previous experiment. I have already incorporated these revisions into the updated results.

3. The time-series data for all provinces should be visually presented to observe any common distribution patterns among them. The plot of these can show us the outliers and common groups of provinces. These can also be applied to all three data sets. 

o I aim to visualize the surveillance data from our three datasets. Regrettably, when I attempted to plot data from all 77 provinces together, it resulted in a cluttered visualization. Consequently, I've decided to create more visually appealing representations by focusing on specific clusters and comparing them with major provinces in figures 8-10.

4. The authors should explain how they selected the number of clusters in the DBSCAN algorithm and explore different parameters to improve clustering results. The reasons for choosing DBSCAN as the clustering algorithm and determining optimal parameters (e.g., epsilon and minimum points) should be elaborated. 

o I’ve added more explanation about DBSCAN and our fine-tuning parameters in line 215-255.

5. The authors should address the situations or policies that could have contributed to the resulting clusters, such as the behavior of people, travel patterns, or government policies. Practical implications for policymakers and public health officials need to be discussed, explaining how this knowledge can inform decision-making and response strategies during future pandemics or endemic phases. What are the situations or policies behind the resulting clusters? These should be clearly addressed. It might be because of the behavior of the Thai people or because workers always come to Bangkok and go back to their homes during long holidays or a knock-down policy. 

o I’ve added information related to public health policies in line 68-81 and case study in Mexico and Singapore in line 86-92.

6. The authors should compare their findings with previous studies on similar topics to validate their results and strengthen their conclusions. Some other clustering approaches may provide different clustering results. 

o I have attempted to identify similar clusters for H1N1 and COVID-19 in Thailand but haven't found a match yet. Before choosing DBSCAN as a clustering method, I experimented with other techniques such as K-means and Self-Organizing Map (SOM), but unfortunately, they did not yield significant results.

7. Limitations, including inherent biases in clustering analysis and potential missed aspects of disease transmission, should be acknowledged. 

o I’ve added technical limitation in 247-250.

8. All figures should be properly labeled with axis titles, and higher-resolution figures should be provided to improve clarity. It may be not necessary to have Figure 1 in the circumstance of the analysis. Only report the number in the text would be enough. 

o I’ve added the axis labels for all figures and recheck the resolution of figures.

9. There is a lack of discussion part that the author can discuss the methods, results, practical implications, and limitation of their study. 

o I’ve added more discussion and explanation in method part and result part.

---

## [Decision Letter · Decision Letter 1]

22 Dec 2023

Comparison of Geological Clusters between Influenza and COVID-19 in Thailand with Unsupervised Clustering Analysis

PONE-D-23-19116R1

Dear Dr. Ahn,

We’re pleased to inform you that your manuscript has been judged scientifically suitable for publication and will be formally accepted for publication once it meets all outstanding technical requirements.

Kind regards,

Samrat Kumar Dey.

Academic Editor

PLOS ONE

Additional Editor Comments (optional):

All the reviewer's comments have been properly addressed. Some minor observations are still provided by the reviewers (Reviewer 2).

For final submission, address those comments and re-submit the article. However, I am accepting the article at this point.

Reviewers' comments:

Reviewer's Responses to Questions

**Comments to the Author**

1. If the authors have adequately addressed your comments raised in a previous round of review and you feel that this manuscript is now acceptable for publication, you may indicate that here to bypass the “Comments to the Author” section, enter your conflict of interest statement in the “Confidential to Editor” section, and submit your "Accept" recommendation.

Reviewer #1: All comments have been addressed

Reviewer #2: All comments have been addressed

2. Is the manuscript technically sound, and do the data support the conclusions?

Reviewer #1: Yes

Reviewer #2: Partly

3. Has the statistical analysis been performed appropriately and rigorously? 

Reviewer #1: Yes

Reviewer #2: Yes

4. Have the authors made all data underlying the findings in their manuscript fully available?

Reviewer #1: Yes

Reviewer #2: Yes

5. Is the manuscript presented in an intelligible fashion and written in standard English?

Reviewer #1: Yes

Reviewer #2: Yes

6. Review Comments to the Author

Reviewer #1: Thank you for addressing all the comments; the revisions greatly enhance the manuscript. Your efforts are appreciated.

Reviewer #2: More information and discussion are all addressed.

However, all figures still have low resolution. Please provide nicer and clearer font size and sharp resolutions for the text in the figures.

7. PLOS authors have the option to publish the peer review history of their article (what does this mean?). If published, this will include your full peer review and any attached files.

Reviewer #1: No

Reviewer #2: No

---

## [Editor Report · Acceptance letter]

12 Jan 2024

PONE-D-23-19116R1 

PLOS ONE

Dear Dr. Ahn, 

I'm pleased to inform you that your manuscript has been deemed suitable for publication in PLOS ONE. Congratulations! Your manuscript is now being handed over to our production team.

Kind regards, 

on behalf of

Mr. Samrat Kumar Dey 

Academic Editor

PLOS ONE